# Understanding and Improving Transformer From a Multi-Particle Dynamic System Point of View

## Abstract

The Transformer architecture is widely used in natural language processing. Despite its success, the design principle of the Transformer remains elusive. In this paper, we provide a novel perspective towards understanding the architecture: we show that the Transformer can be mathematically interpreted as a *numerical Ordinary Differential Equation (ODE) solver for a convection-diffusion equation in a multi-particle dynamic system*. In particular, how words in a sentence are abstracted into contexts by passing through the layers of the Transformer can be interpreted as approximating multiple particles' movement in the space using the Lie-Trotter splitting scheme and the Euler's method. Given this ODE's perspective, the rich literature of numerical analysis can be brought to guide us in designing effective structures beyond the Transformer. As an example, we propose to replace the Lie-Trotter splitting scheme by the Strang-Marchuk splitting scheme, a scheme that is more commonly used and with much lower local truncation errors. The Strang-Marchuk splitting scheme suggests that the self-attention and position-wise feed-forward network (FFN) sub-layers should not be treated equally. Instead, in each layer, two position-wise FFN sub-layers should be used, and the self-attention sub-layer is placed in between. This leads to a brand new architecture. Such an FFN-attention-FFN layer is "Macaron-like", and thus we call the network with this new architecture the *Macaron* Net. Through extensive experiments, we show that the Macaron Net is superior to the Transformer on both supervised and unsupervised learning tasks. The reproducible code can be found on http://anonymized

## 1 Introduction

The Transformer is one of the most commonly used neural network architectures in natural language processing. Variants of the Transformer have achieved state-of-the-art performance in many tasks including language modeling (Dai et al., 2019; Al-Rfou et al., 2018) and machine translation (Vaswani et al., 2017; Dehghani et al., 2018; Edunov et al., 2018). Unsupervised pre-trained models based on the Transformer architecture also show impressive performance in many downstream tasks (Radford et al., 2019; Devlin et al., 2018).

The Transformer architecture is mainly built by stacking layers, each of which consists of two sub-layers with residual connections: the self-attention sub-layer and the position-wise feed-forward network (FFN) sub-layer. For a given sentence, the self-attention sub-layer considers the semantics and dependencies of words at different positions and uses that information to capture the internal structure and representations of the sentence. The position-wise FFN sub-layer is applied to each position separately and identically to encode context at each position into higher-level representations. Although the Transformer architecture has demonstrated promising results in many tasks, its design principle is not fully understood, and thus the strength of the architecture is not fully exploited. As far as we know, there is little work studying the foundation of the Transformer or different design choices.

In this paper, we provide a novel perspective towards understanding the architecture. In particular, we are the first to show that the Transformer architecture is inherently related to the Multi-Particle

Dynamic System (MPDS) in physics. MPDS is a well-established research field which aims at modeling how a collection of particles move in the space using differential equations (Moulton, 2012).

In MPDS, the behavior of each particle is usually modeled by two factors separately. The first factor is the convection which concerns the mechanism of each particle regardless of other particles in the system, and the second factor is the diffusion which models the movement of the particle resulting from other particles in the system.

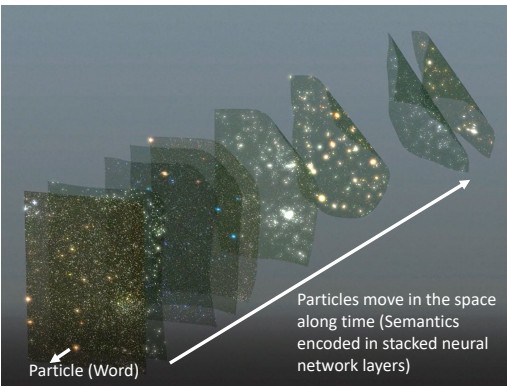

Inspired by the relationship between the ODE and neural networks (Lu et al., 2017; Chen et al., 2018a), we first show that the Transformer layers can be naturally interpreted as a numerical ODE solver for a first-order convection-diffusion equation in MPDS. To be more specific, the self-attention sub-layer, which transforms the semantics at one position by attending over all other

Figure 1: Physical interpretation of Transformer.

positions, corresponds to the diffusion term; The position-wise FFN sub-layer, which is applied to each position separately and identically, corresponds to the convection term. The number of stacked layers in the Transformer corresponds to the time dimension in ODE. In this way, the stack of self-attention sub-layers and position-wise FFN sub-layers with residual connections can be viewed as solving the ODE problem numerically using the Lie-Trotter splitting scheme (Geiser, 2009) and the Euler's method (Ascher & Petzold, 1998). By this interpretation, we have a novel understanding of learning contextual representations of a sentence using the Transformer: the feature (a.k.a, embedding) of words in a sequence can be considered as the initial positions of a collection of particles, and the latent representations abstracted in stacked Transformer layers can be viewed as the location of particles moving in a high-dimensional space at different time points.

Such an interpretation not only provides a new perspective on the Transformer but also inspires us to design new structures by leveraging the rich literature of numerical analysis. The Lie-Trotter splitting scheme is simple but not accurate and often leads to high approximation error (Geiser, 2009). The Strang-Marchuk splitting scheme (Strang, 1968) is developed to reduce the approximation error by a simple modification to the Lie-Trotter splitting scheme and is theoretically more accurate. Mapped to neural network design, the Strang-Marchuk splitting scheme suggests that there should be three sub-layers: two position-wise feed-forward sub-layers with half-step residual connections and one self-attention sub-layer placed in between with a full-step residual connection. By doing so, the stacked layers will be more accurate from the ODE's perspective and will lead to better performance in deep learning. As the FFN-attention-FFN layer is "Macaron-like", we call it *Macaron layer* and call the network composed of Macaron layers the *Macaron Net*.

We conduct extensive experiments on both supervised and unsupervised learning tasks. For each task, we replace Transformer layers by Macaron layers and keep the number of parameters to be the same. Experiments show that the Macaron Net can achieve higher accuracy than the Transformer on all tasks which, in a way, is consistent with the ODE theory.

## 2 BACKGROUND

### 2.1 RELATIONSHIP BETWEEN NEURAL NETWORKS AND ODE

Recently, there are extensive studies to bridge deep neural networks with ordinary differential equations (Weinan, 2017; Lu et al., 2017; Haber & Ruthotto, 2017; Chen et al., 2018a; Zhang et al., 2019b; Sonoda & Murata, 2019; Thorpe & van Gennip, 2018). We here present a brief introduction to such a relationship and discuss how previous works borrow powerful tools from numerical analysis to help deep neural network design.

A first-order ODE problem is usually defined as to solve the equation (i.e., calculate $x(t)$ for any $t$) which satisfies the following first-order derivative and the initial condition:

$$\frac{\mathrm{d}x(t)}{\mathrm{d}t} = f(x, t), \quad x(t_0) = w, \tag{1}$$

in which $x(t) \in \mathbb{R}^d$ for all $t \geq t_0$. ODEs usually have physical interpretations. For example, $x(t)$ can be considered as the location of a particle moving in the $d$-dimensional space and the first order time derivative can be considered as the velocity of the particle.

Usually there is no analytic solution to Eqn (1) and the problem has to be solved numerically. The simplest numerical ODE solver is the Euler's method (Ascher & Petzold, 1998). The Euler's method discretizes the time derivative $\frac{\mathrm{d}x(t)}{\mathrm{d}t}$ by its first-order approximation $\frac{x(t_2)-x(t_1)}{t_2-t_1} \approx f(x(t_1), t_1)$. By doing so, for the fixed time horizon $T = t_0 + \gamma L$, we can estimate $x(T)$ from $x_0 \doteq x(t_0)$ by sequentially estimating $x_{l+1} \doteq x(t_{l+1})$ using

$$x_{l+1} = x_l + \gamma f(x_l, t_l) \tag{2}$$

where $l = 0, \cdots, L-1$, $t_l = t_0 + \gamma l$ is the time point corresponds to $x_l$, and $\gamma = (T - t_0)/L$ is the step size. As we can see, this is mathematically equivalent to the ResNet architecture (Lu et al., 2017; Chen et al., 2018a): The function $\gamma f(x_l, t_l)$ can be considered as a neural-network block, and the second argument $t_l$ in the function indicates the set of parameters in the $l$-th layer. The simple temporal discretization by Euler's method naturally leads to the residual connection.

Observing such a strong relationship, researchers use ODE theory to explain and improve the neural network architectures mainly designed for computer vision tasks. Lu et al. (2017); Chen et al. (2018a) show any parametric ODE solver can be viewed as a deep residual network (probably with infinite layers), and the parameters in the ODE can be optimized through backpropagation. Recent works discover that new neural networks inspired by sophisticated numerical ODE solvers can lead to better performance. For example, Zhu et al. (2018) uses a high-precision Runge-Kutta method to design a neural network, and the new architecture achieves higher accuracy. Haber & Ruthotto (2017) uses a leap-frog method to construct a reversible neural network. Liao & Poggio (2016); Chang et al. (2019) try to understand recurrent neural networks from the ODE's perspective, and Tao et al. (2018) uses non-local differential equations to model non-local neural networks.

## 2.2 TRANSFORMER

The Transformer architecture is usually developed by stacking Transformer layers (Vaswani et al., 2017; Devlin et al., 2018). A Transformer layer operates on a sequence of vectors and outputs a new sequence of the same shape. The computation inside a layer is decomposed into two steps: the vectors first pass through a (multi-head) self-attention sub-layer and the output will be further put into a position-wise feed-forward network sub-layer. Residual connection (He et al., 2016) and layer normalization (Lei Ba et al., 2016) are employed for both sub-layers. The visualization of a Transformer layer is shown in Figure 2(a) and the two sub-layers are defined as below.

**Self-attention sub-layer** The attention mechanism can be formulated as querying a dictionary with key-value pairs (Vaswani et al., 2017), e.g., $\text{Attention}(Q, K, V) = \text{softmax}(QK^T/\sqrt{d_{model}}) \cdot V$, where $d_{model}$ is the dimensionality of the hidden representations and $Q$ (Query), $K$ (Key), $V$ (Value) are specified as the hidden representations of the previous layer in the so-called *self-attention* sub-layers in the Transformer architecture. The multi-head variant of attention allows the model to jointly attend to information from different representation subspaces, and is defined as

$$\text{Multi-head}(Q, K, V) = \text{Concat}(\text{head}_1, \cdots, \text{head}_H)W^O, \tag{3}$$

$$\text{head}_k = \text{Attention}(QW_k^Q, KW_k^K, VW_k^V), \tag{4}$$

where $W_k^Q \in \mathbb{R}^{d_{model} \times d_K}, W_k^K \in \mathbb{R}^{d_{model} \times d_K}, W_k^V \in \mathbb{R}^{d_{model} \times d_V}$, and $W^O \in \mathbb{R}^{Hd_V \times d_{model}}$ are project parameter matrices, $H$ is the number of heads, and $d_K$ and $d_V$ are the dimensionalities of Key and Value.

**Position-wise FFN sub-layer** In addition to the self-attention sub-layer, each Transformer layer also contains a fully connected feed-forward network, which is applied to each position separately and identically. This feed-forward network consists of two linear transformations with an activation function $\sigma$ in between. Specially, given vectors $h_1, \ldots, h_n$, a position-wise FFN sub-layer transforms each $h_i$ as $\text{FFN}(h_i) = \sigma(h_i W_1 + b_1)W_2 + b_2$, where $W_1, W_2, b_1$ and $b_2$ are parameters.

In this paper, we take the first attempt to provide an understanding of the feature extraction process in natural language processing from the ODE's viewpoint. As discussed in Section 2.1, several

works interpret the standard ResNet using the ODE theory. However, we found this interpretation cannot be directly applied to the Transformer architecture. First, different from vision applications whose size of the input (e.g., an image) is usually predefined and fixed, the input (e.g., a sentence) in natural language processing is always of variable length, which makes the single-particle ODE formulation used in previous works not applicable. Second, the Transformer layer contains very distinct sub-layers. The self-attention sub-layer takes the information from all positions as input while the position-wise feed-forward layer is applied to each position separately. How to interpret these heterogeneous components by ODE is also not covered by previous works (Tao et al., 2018; Chen et al., 2018a).

## 3 REFORMULATE TRANSFORMER LAYERS AS AN ODE SOLVER FOR MULTI-PARTICLE DYNAMIC SYSTEM

In this section, we first introduce the general form of differential equations in MPDS and then reformulate the stacked Transformer layers to show they form a numerical ODE solver for a specific problem. After that, we use advanced methods in the ODE theory to design new architectures.

### 3.1 MULTI-PARTICLE ODE AND ITS NUMERICAL SOLVER

Understanding the dynamics of multiple particles' movements in space is one of the important problems in physics, especially in fluid mechanics and astrophysics (Moulton, 2012). The behavior of each particle is usually modeled by two factors: The first factor concerns about the mechanism of its movement regardless of other particles, e.g., caused by an external force outside of the system, which is usually referred to as the convection; The second factor concerns about the movement resulting from other particles, which is usually referred to as the diffusion. Mathematically, assume there are $n$ particles in $d$-dimensional space. Denote $x_i(t) \in \mathbb{R}^d$ as the location of $i$-th particle at time $t$. The dynamics of particle $i$ can be formulated as

$$
\begin{aligned}
\frac{\mathrm{d}x_i(t)}{\mathrm{d}t} &= F(x_i(t), [x_1(t), \cdots, x_n(t)], t) + G(x_i(t), t), \\
x_i(t_0) &= w_i, \quad i = 1, \ldots, n.
\end{aligned}
\tag{5}
$$

Function $F(x_i(t), [x_1(t), \cdots, x_n(t)], t)$ represents the diffusion term which characterizes the interaction between the particles. $G(x, t)$ is a function which takes a location $x$ and time $t$ as input and represents the convection term.

**Splitting schemes**   As we can see, there are two coupled terms in the right-hand side of Eqn (5) describing different physical phenomena. Numerical methods of directly solving such ODEs can be complicated. The splitting method is a prevailing way of solving such coupled differential equations that can be decomposed into a sum of differential operators (McLachlan & Quispel, 2002). Furthermore, splitting convection from diffusion is quite standard for many convection-diffusion equations (Glowinski et al., 2017; Geiser, 2009). The Lie-Trotter splitting scheme (Geiser, 2009) is the simplest splitting method. It splits the right-hand side of Eqn (5) into function $F(\cdot)$ and $G(\cdot)$ and solves the individual dynamics alternatively. More precisely, to compute $x_i(t + \gamma)$ from $x_i(t)$, the Lie-Trotter splitting scheme with the Euler's method reads as

$$
\tilde{x}_i(t) = x_i(t) + \gamma F(x_i(t), [x_1(t), x_2(t), \cdots, x_n(t)], t),
\tag{6}
$$

$$
x_i(t + \gamma) = \tilde{x}_i(t) + \gamma G(\tilde{x}_i(t), t).
\tag{7}
$$

From time $t$ to time $t + \gamma$, the Lie-Trotter splitting method first solves the ODE with respect to $F(\cdot)$ and acquire an intermediate location $\tilde{x}_i(t)$. Then, starting from $\tilde{x}_i(t)$, it solves the second ODE with respect to $G(\cdot)$ to obtain $x_i(t + \gamma)$.

### 3.2 PHYSICAL INTERPRETATION OF THE TRANSFORMER

We reformulate the two sub-layers of the Transformer in order to match its form with the ODE described above. Denote $x_l = (x_{l,1}, \ldots, x_{l,n})$ as the input to the $l$-th Transformer layer, where $n$ is the sequence length and $x_{l,i}$ is a real-valued vector in $\mathbb{R}^d$ for any $i$.

**Reformulation of the self-attention sub-layer**  Denote $\tilde{x}_{l,i}$ as the output of the (multi-head) self-attention sub-layer at position $i$ with residual connections. The computation of $\tilde{x}_{l,i}$ can be written as

$$\tilde{x}_{l,i} = x_{l,i} + \text{Concat}\left(\text{head}_1, ..., \text{head}_H\right) W^{O,l}, \tag{8}$$

$$\text{where head}_k = \sum_{j=1}^{n} \alpha_{ij}^{(k)} [x_{l,j} W_k^{V,l}] = \sum_{j=1}^{n} \left( \frac{\exp(e_{ij}^{(k)})}{\sum_{q=1}^{n} \exp(e_{iq}^{(k)})} \right) [x_{l,j} W_k^{V,l}], \tag{9}$$

and $e_{ij}^{(k)}$ is computed as the dot product of input $x_{l,i}$ and $x_{l,j}$ with linear projection matrices $W_k^{Q,l}$ and $W_k^{K,l}$, i.e., $e_{ij}^{(k)} = d_{model}^{-1/2} \cdot (x_{l,i} W_k^{Q,l})(x_{l,j} W_k^{K,l})^T$. Considering $\alpha_{ij}^{(k)}$ as a normalized value of the pair-wise dot product $e_{ij}^{(k)}$ over $j$, we can generally reformulate Eqn (8) as

$$\tilde{x}_{l,i} = x_{l,i} + \text{MultiHeadAtt}_{W_{att}^l}(x_{l,i}, [x_{l,1}, x_{l,2}, \cdots, x_{l,n}]), \tag{10}$$

where $W_{att}^l$ denotes all trainable parameters in the $l$-th self-attention sub-layer.

**Reformulation of the position-wise FFN sub-layer**  Next, $\tilde{x}_{l,i}$ is put into the position-wise FFN sub-layer with residual connections and output $x_{l+1,i}$. The computation of $x_{l+1,i}$ can be written as

$$x_{l+1,i} = \tilde{x}_{l,i} + \text{FFN}_{W_{ffn}^l}(\tilde{x}_{l,i}), \tag{11}$$

where $W_{ffn}^l$ denotes all trainable parameters in the $l$-th position-wise FFN sub-layer.

**Reformulation of Transformer layers**  Combining Eqn (10) and (11), we reformulate the Transformer layers[1] as

$$\tilde{x}_{l,i} = x_{l,i} + \text{MultiHeadAtt}_{W_{att}^l}(x_{l,i}, [x_{l,1}, x_{l,2}, \cdots, x_{l,n}]), \tag{12}$$

$$x_{l+1,i} = \tilde{x}_{l,i} + \text{FFN}_{W_{ffn}^l}(\tilde{x}_{l,i}). \tag{13}$$

We can see that the Transformer layers (Eqn (12-13)) resemble the multi-particle ODE solver in Section 3.1 (Eqn (6-7)). Indeed, we can formally establish the link between the ODE solver with splitting scheme and stacked Transformer layers as below.

**Claim 1.** *Define* $\gamma F^*(x_{l,i}, [x_{l,1}, \cdots, x_{l,n}], t_l) = MultiHeadAtt_{W_{att}^l}(x_{l,i}, [x_{l,1}, \cdots, x_{l,n}])$ *and* $\gamma G^*(x_{l,i}, t_l) = FFN_{W_{ffn}^l}(x_{l,i})$. *The Transformer can be viewed as a numerical ODE solver using Lie-Trotter splitting scheme and the Euler's method (with time step $\gamma$) for Eqn (5) with $F^*$ and $G^*$.*

The above observation grants a physical interpretation of natural language processing and provides a new perspective on the Transformer architecture. First, this perspective provides a unified view of the heterogeneous components in the Transformer. The self-attention sub-layer is viewed as a diffusion term which characterizes the particle interactions while the position-wise feed-forward network sub-layer is viewed as a convection term. The two terms together naturally form the convection-diffusion equation in physics. Second, this interpretation advances our understanding of the latent representations of language through the Transformer. Viewing the feature (a.k.a., embedding) of words in a sequence as the initial position of particles, we can interpret the latent representations of the sentence abstracted by the Transformer as particles moving in a high-dimensional space as demonstrated in Figure 1 (Zhang, 2019).

## 3.3 Improving Transformer Via Strang-Marchuk Splitting Scheme

In the previous subsection, we have successfully mapped the Transformer architecture to a numerical ODE solver for MPDS. However, we would like to point out that one of the key components in this ODE solver, the Lie-Trotter splitting scheme, is the simplest one but has relatively high errors. In this

---

[1]Layer normalization is sometimes applied to the sub-layers but recent work (Zhang et al., 2019a) shows that the normalization trick is not essential and can be removed. One can still readily check that the reformulation (Eqn (12) and (13)) still holds with layer normalization.

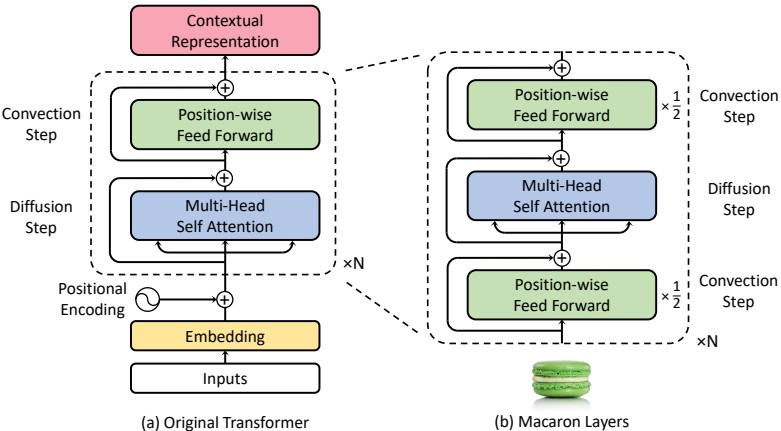

Figure 2: The Transformer and our Macaron architectures.

subsection, we incorporate one of the most popular and widely used splitting scheme (Geiser, 2009), the Strang-Marchuk splitting scheme, into the design of the neural networks.

The Lie-Trotter splitting scheme solves the dynamics of $F(\cdot)$ and $G(\cdot)$ alternatively and exclusively in that order. This inevitably brings bias and leads to higher local truncation errors (Geiser, 2009). To mitigate the bias, we use a simple modification to the Lie-Trotter splitting scheme by dividing the one-step numerical solver for $G(\cdot)$ into two *half-steps*: we put one half-step before solving $F(\cdot)$ and put the other half-step after solving $F(\cdot)$. This modified splitting scheme is known as the Strang-Marchuk splitting scheme (Strang, 1968). Mathematically, to compute $x_i(t + \gamma)$ from $x_i(t)$, the Strang-Marchuk splitting scheme reads as

$$\tilde{x}_i(t) = x_i(t) + \frac{\gamma}{2}G(x_i(t), t), \tag{14}$$

$$\hat{x}_i(t) = \tilde{x}_i(t) + \gamma F(\tilde{x}_i(t), [\tilde{x}_1(t), \tilde{x}_2(t), \cdots, \tilde{x}_n(t)], t), \tag{15}$$

$$x_i(t + \gamma) = \hat{x}_i(t) + \frac{\gamma}{2}G\left(\hat{x}_i(t), t + \frac{\gamma}{2}\right). \tag{16}$$

The Strang-Marchuk splitting scheme enjoys higher-order accuracy than the Lie-Trotter splitting scheme (Bobylev & Ohwada, 2001) in terms of the *local truncation error* (Ascher & Petzold, 1998), which measures the per-step distance between the true solution and the approximated solution using numerical schemes. Mathematically, for a differential equation $\frac{dx(t)}{dt} = f(x, t)$ and a numerical scheme $\mathcal{A}$, the local truncation error of numerical scheme $\mathcal{A}$ is defined as $\tau = x(t + \gamma) - \mathcal{A}(x(t), \gamma)$. For example, when $\mathcal{A}$ is the Euler's method, $\tau_{Euler} = x(t + \gamma) - x(t) - \gamma f(x(t), t)$. The order of local truncation error of the two schemes has been studied in Bobylev & Ohwada (2001), as shown in the following theorem.

**Theorem 1.** *(Bobylev & Ohwada, 2001) The local truncation error of the Lie-Trotter splitting scheme is **second-order** ($O(\gamma^2)$) and the local truncation error of the Strang-Marchuk splitting scheme is **third-order** ($O(\gamma^3)$).*

For completeness, we provide the formal theorem with proof in the appendix. We can see from Eqn (14-16) that the Strang-Marchuk splitting scheme uses a three-step process to solve the ODE. Mapped to neural network design, the Strang-Marchuk splitting scheme (together with the Euler's method) suggests there should also be three sub-layers instead of the two sub-layers in the Transformer. By replacing function $\gamma F$ and $\gamma G$ by MultiHeadAtt and FFN, we have

$$\tilde{x}_{l,i} = x_{l,i} + \frac{1}{2}\text{FFN}_{W_{ffn}^{l, down}}(x_{l,i}), \tag{17}$$

$$\hat{x}_{l,i} = \tilde{x}_{l,i} + \text{MultiHeadAtt}_{W_{att}^l}(\tilde{x}_{l,i}, [\tilde{x}_{l,1}, \tilde{x}_{l,2}, \cdots, \tilde{x}_{l,n}]), \tag{18}$$

$$x_{l+1,i} = \hat{x}_{l,i} + \frac{1}{2}\text{FFN}_{W_{ffn}^{l, up}}(\hat{x}_{l,i}). \tag{19}$$

Table 1: Translation performance (BLEU) on IWSLT14 De-En and WMT14 En-De testsets.

| Method | IWSLT14 De-En `small` | WMT14 En-De `base` | `big` |
|---|---|---|---|
| Transformer (Vaswani et al., 2017) | 34.4 | 27.3 | 28.4 |
| Weighted Transformer (Ahmed et al., 2017) | / | 28.4 | 28.9 |
| Relative Transformer (Shaw et al., 2018) | / | 26.8 | 29.2 |
| Universal Transformer (Dehghani et al., 2018) | / | **28.9** | / |
| Scaling NMT (Ott et al., 2018) | / | / | 29.3 |
| Dynamic Conv (Wu et al., 2019a) | 35.2 | / | 29.7 |
| **Macaron Net** | **35.4** | **28.9** | **30.2** |

From Eqn (17-19), we can see that the new layer composes of three sub-layers. Each hidden vector at different positions will first pass through the first position-wise FFN sub-layer with a half-step [2] residual connection ("$\frac{1}{2}$" in Eqn (17)), and then the output vectors will be feed into a self-attention sub-layer. In the last step, the output vectors from the self-attention sub-layer will be put into the second position-wise FFN sub-layer with a half-step residual connection. Since the FFN-attention-FFN structure is "Macaron"-like, we call the layer as *Macaron* layer and call the network using Macaron layers as Macaron Net, as shown in Figure 2(b). Previous works (Lu et al., 2017; Zhu et al., 2018) have successfully demonstrated that the neural network architectures inspired by higher-order accurate numerical ODE solvers will lead to better results in deep learning and we believe the Macaron Net can achieve better performance on practical natural language processing applications than the Transformer.

## 4 EXPERIMENTS

We test our proposed Macaron architectures in both supervised and unsupervised learning setting. For the supervised learning setting, we use IWLST14 and WMT14 machine translation datasets. For the unsupervised learning setting, we pretrain the model using the same method as in Devlin et al. (2018) and test the learned model over a set of downstream tasks. Extra descriptions about datasets, model specifications, and hyperparameter configurations can be found in the appendix.

### 4.1 EXPERIMENT SETTINGS

**Machine Translation** Machine translation is an important application for natural language processing (Vaswani et al., 2017). We evaluate our methods on two widely used public datasets: IWSLT14 German-to-English (De-En) and WMT14 English-to-German (En-De) dataset.

For the WMT14 dataset, the basic configurations of the Transformer architecture are the `base` and the `big` settings (Vaswani et al., 2017). Both of them consist of a 6-layer encoder and 6-layer decoder. The size of the hidden nodes and embeddings are set to 512 for `base` and 1024 for `big`. The number of heads are 8 for `base` and 16 for `big`. Since the IWSLT14 dataset is much smaller than the WMT14 dataset, the `small` setting is usually used, whose size of hidden states and embeddings is set to 512 and the number of heads is set to 4. For all settings, the dimensionality of the inner-layer of the position-wise FFN is four times of the dimensionality of the hidden states.

For each setting (`base`, `big` and `small`), we replace all Transformer layers by the Macaron layers[3] and obtain the `base`, `big` and `small` Macaron, each of which contains two position-wise feed-

---

[2]The half-step is critical in the Strang-Marchuk splitting scheme to solve an ODE problem, but it may be not essential in training a particular neural network. For example, the FFN sub-layer in the Transformer is designed as $FFN(h_i) = \sigma(h_i W_1 + b_1)W_2 + b_2$. Placing $1/2$ to rescale this FFN sub-layer is equivalent to rescale the parameter $W_2$ and $b_2$ from the beginning of the optimization.

[3]The translation model is based on the encoder-decoder framework. In the Transformer, the decoder layer has a third sub-layer which performs multi-head attention over the output of the encoder stack (encoder-decoder-attention) and a mask to prevent positions from attending to subsequent positions. In our implementation of Macaron decoder, we also use masks and split the FFN into two sub-layers and thus our decoder layer is (FFN, self-attention, encoder-decoder-attention, and FFN).

Table 2: Test results on the GLUE benchmark (except WNLI).

| Method | CoLA | SST-2 | MRPC | STS-B | QQP | MNLI-m/mm | QNLI | RTE | GLUE |
|---|---|---|---|---|---|---|---|---|---|
| *Existing systems* | | | | | | | | | |
| ELMo (Peters et al., 2018) | 33.6 | 90.4 | 84.4/78.0 | 74.2/72.3 | 63.1/84.3 | 74.1/74.5 | 79.8 | 58.9 | 70.0 |
| OpenAI GPT (Radford et al.) | 47.2 | 93.1 | 87.7/83.7 | 85.3/84.8 | 70.1/88.1 | 80.7/80.6 | 87.2 | 69.1 | 76.9 |
| BERT base (Devlin et al., 2018) | 52.1 | 93.5 | **88.9/84.8** | 87.1/85.8 | **71.2/89.2** | 84.6/83.4 | 90.5 | 66.4 | 78.3 |
| *Our systems* | | | | | | | | | |
| BERT base (ours) | 52.8 | 92.8 | 87.3/83.0 | 81.2/80.0 | 70.2/88.4 | 84.4/83.7 | 90.4 | 64.9 | 77.4 |
| Macaron Net base | **57.6** | **94.0** | 88.4/84.4 | **87.5/86.3** | 70.8/89.0 | **85.4/84.5** | **91.6** | **70.5** | **79.7** |

forward sub-layers in a layer. To make a fair comparison, we set the dimensionality of the inner-layer of the two FFN sub-layers in the Macaron layers to two times of the dimensionality of the hidden states. By doing this, the base, big and small Macaron have the same number of parameters as the base, big and small Transformer respectively.

**Unsupervised Pretraining** BERT (Devlin et al., 2018) is the current state-of-the-art pre-trained contextual representation model based on a multi-layer Transformer encoder architecture and trained by masked language modeling and next-sentence prediction tasks. We compare our proposed Macaron Net with the base setting from the original paper (Devlin et al., 2018), which consists of 12 Transformer layers. The size of hidden states and embeddings are set to 768, and the number of attention heads is set to 12. Similarly, we replace the Transformer layers in BERT base by the Macaron layers and reduce the dimensionality of the inner-layer of the two FFN sub-layers by half, and thus we keep the number of parameters of our Macaron base same as BERT base.

## 4.2 EXPERIMENT RESULTS

**Machine Translation** We use BLEU (Papineni et al., 2002) as the evaluation measure for machine translation. Following common practice, we use tokenized case-sensitive BLEU and case-insensitive BLEU for WMT14 En-De and IWSLT14 De-En respectively.

The results for machine translation are shown in Table 1. For the IWSLT14 dataset, our Macaron small outperforms the Transformer small by 1.0 in terms of BLEU. For the WMT14 dataset, our Macaron base outperforms its Transformer counterpart by 1.6 BLEU points. Furthermore, the performance of our Macaron base model is even better than that of the Transformer big model reported in Vaswani et al. (2017), but with much less number of parameters. Our Macaron big outperforms the Transformer big by 1.8 BLEU points. Comparing with other concurrent works, the improvements in our proposed method are still significant.

**Unsupervised Pretraining** Following Devlin et al. (2018), we evaluate all models by fine-tuning them on 8 downstream tasks in the General Language Understanding Evaluation (GLUE) benchmark (Wang et al., 2019), including CoLA (Warstadt et al., 2018), SST-2 (Socher et al., 2013), MRPC (Dolan & Brockett, 2005), STS-B (Cer et al., 2017), QQP (Chen et al., 2018b), MNLI (Williams et al., 2018), QNLI (Rajpurkar et al., 2016), and RTE (Bentivogli et al., 2009). More details about individual tasks and their evaluation metrics can be found in the appendix and Wang et al. (2019); Devlin et al. (2018). To fine-tune the models, we follow the hyperparameter search space in Devlin et al. (2018) for all downstream tasks, including different batch sizes, learning rates, and numbers of epochs.

The GLUE results are presented in Table 2. We present the results of two BERT base models. One is from Devlin et al. (2018), and the other is trained using our own data. Due to that some pieces of data used in Devlin et al. (2018) are no longer freely distributed, the two numbers are slightly different. We can see from the table, our proposed Macaron Net base outperforms all baselines in terms of the general GLUE score. Specifically, given the same dataset, our proposed model outperforms our trained BERT base model in all tasks. Even comparing with the BERT base model in Devlin et al. (2018), our model performs better in 6 out of 8 tasks and achieves close performance in the rest 2 tasks.

Table 3: Additional Comparisions on WMT14 En-De translation tasks.

| Method | #Params | BLEU |
|---|---|---|
| Att-FFN-Att | 213M | 28.6 |
| No residual | 213M | 28.3 |
| 8-Layer Transformer (`big`, Wu et al., 2019b) | 284M | 28.8 |
| 10-Layer Transformer (`big`, Wu et al., 2019b) | 355M | 28.6 |
| **Macaron Net** (`big`) | 213M | **30.2** |

As a summary, the improvement in both machine translation and GLUE tasks well aligns with the ODE theory and our proposed architecture performs better than the Transformer in real practice.

## 4.3 ADDITIONAL COMPARISONS

As we can see, the main difference between Macaron Net and the Transformer is that Macaron Net uses two FFN sub-layers in a layer block while the Transformer just uses one. One may argue that the improvements of the new architecture may be simply due to adding nonlinearities but not from ODE theory. We point out here that adding nonlinearities does not always guarantee improvements in deep learning. For example, feed-forward neural networks contain much more nonlinearities than convolutional neural networks but the performance of feed-forward neural networks is superior than that of convolutional neural networks in image classification. Furthermore, Wu et al. (2019b) shows that simply adding nonlinearities by stacking more Transformer layers does not work well in practice.

To evaluate the effectiveness of our network, we further conducted experiments to show the ODE-inspired way of adding nonlinearities is better than following heuristic methods:

- *Att-FFN-Att*: In Macaron Net, we use two FFN sub-layers and one attention sub-layer in a layer. Here we construct a baseline that uses two attention sub-layers and one FFN sub-layer layer. Note that the attention sub-layer also contains nonlinearities in the softmax and dot-product operations.
- *No residual*: In the Transformer, the FFN sub-layer has one hidden layer. We changed it to two hidden layers without residual connections, which increases the nonlinearities.

We compare these two models with Macaron Net on WMT14 En-De task and keep all the model parameters to be the same as the Transformer `big` setting. We list the performance of different models in Table 3. We can see that the BLEU scores of two models are 28.6/28.3 respectively. As a comparison, the BLEU score of Macaron Net is 30.2. We also cite the results from Wu et al. (2019b), which shows that simply stacking more Transformer layers cannot reach comparable performance. Therefore, we believe our ODE-inspired architecture design is principled and better than heuristics, which provides a better understanding to the Transformer model.

Furthermore, we can also see that the empirical results are consistent with the ODE theories. The step size $\gamma$ in ODE relates to #layers in deep learning. In detail, a small step size $\gamma$ maps to a large value of #layer: a neural network with more layers stacked corresponds to an ODE solver with smaller step size $\gamma$. In Table 1 and 2, we can see that that given the same step size $\gamma$ (#layers), our Macaron Net is better than the Transformer. Results in Wu et al. (2019b) show that even using a smaller $\gamma$ (8-layer or 10-layer Transformer), our Macaron Net (6-layer) is still better. These results are consistent with ODE theories: (1) A higher-order ODE solver is better given the same step size (6-layer Macaron Net v.s. 6-layer Transformer). (2) The order matters, a higher-order ODE solver works well even using a relatively larger step size (6-layer Macaron Net v.s. 10-layer Transformer).

## 5 CONCLUSION AND FUTURE WORK

In this paper, we interpret the Transformer as a numerical ODE solver for a convection-diffusion equation in a multi-particle dynamic system. Specifically, how words in a sentence are abstracted into contexts by passing through the layers of the Transformer can be interpreted as approximating multiple particles' movement in the space using the Lie-Trotter splitting scheme and the Euler's method. By replacing the Lie-Trotter splitting scheme with the more advanced Strang-Marchuk

splitting scheme, we obtain a new architecture. The improvements in real applications show the effectiveness of the new model and are consistent with the ODE theory. In the future, we will explore deeper connections between the ODE theory and the Transformer models and use the ODE theory to improve the individual components in the Transformer architecture such as attention modules.

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

APPENDIX

## A   PROOF OF THE THEOREM

**Theorem 2.** *(Bobylev & Ohwada, 2001) We denote the true solution of equation $\frac{\mathrm{d}x}{\mathrm{d}t} = F(x), x(0) = y_0$[4] at time $t$ as $x(t) = S_F^t(y_0)$. Simliarily, we can define $S_G^t$ and $S_{F+G}^t$. The local truncation error at $t = 0$ of the Lie-Trotter splitting scheme is $S_{F+G}^\gamma(y_0) - S_G^\gamma[S_F^\gamma(y_0)] = \frac{\gamma^2}{2}\{F'(y_0)(y_0)G(y_0) - G'(y_0)(y_0)F(y_0)\} + O(\gamma^3)$ which is **second-order**. The local truncation error at $t = 0$ of the Strang-Marchuk splitting scheme $S_G^{\gamma/2}\{S_F^\gamma[S_G^{\gamma/2}(y_0)]\}$ is $S_{F+G}^\gamma(y_0) - S_G^{\gamma/2}\{S_F^\gamma[S_G^{\gamma/2}(y_0)]\} = O(\gamma^3)$ which is **third-order**.*

*Proof.* Since

$$S_F^\gamma(y_0) = y_0 + \gamma F(y_0) + \frac{\gamma^2}{2}F'(y_0)F(y_0) + O(\gamma^3),$$

$$S_G^\gamma(y_0) = y_0 + \gamma G(y_0) + \frac{\gamma^2}{2}G'(y_0)G(y_0) + O(\gamma^3),$$

we have

$$S_G^\gamma[S_F^\gamma(y_0)] = S_F^\gamma(y_0) + \gamma G(S_F^\gamma(y_0)) + \frac{\gamma^2}{2}G'(S_F^\gamma(y_0))G(S_F^\gamma(y_0)) + O(\gamma^3).$$

At the same time, we have

$$G(y_0 + \gamma F(y_0) + O(\gamma^2)) = G(y_0) + \gamma G'(y_0)F(y_0) + O(\gamma^2),$$
$$G'(S_F^\gamma(y_0))G(S_F^\gamma(y_0)) = G'(y_0)G(y_0) + O(\gamma).$$

Combine the estimations we have

$$S_G^\gamma[S_F^\gamma(y_0)] = y_0 + \gamma[F(y_0) + G(y_0)]$$
$$+ \frac{\gamma^2}{2}[G'(y_0)G(y_0) + F'(y_0)F(y_0) + 2G'(y_0)F(y_0)] + O(\gamma^3).$$

As a result, we can estimate the local truncation error of Lie-Trotter splitting scheme as

$$S_{F+G}^\gamma(y_0) - S_G^\gamma[S_F^\gamma(y_0)]$$

$$= y_0 + \gamma(F(y_0) + G(y_0)) + \frac{\gamma^2}{2}(F'(y_0) + G'(y_0))(F(y_0) + G(y_0)) + O(\gamma^3)$$

$$- (y_0 + \gamma[F(y_0) + G(y_0)] + \frac{\gamma^2}{2}[G'(y_0)G(y_0) + F'(y_0)F(y_0) + 2G'(y_0)F(y_0)] + O(\gamma^3))$$

$$= \frac{\gamma^2}{2}\{F'(y_0)G(y_0) - G'(y_0)F(y_0)\} + O(\gamma^3).$$

To estimate the Strang-Marchuk splitting scheme's local truncation error, we rewrite the Strang-Marchuk splitting scheme as

$$S_G^{\gamma/2}\{S_F^\gamma[S_G^{\gamma/2}(y_0)]\} = S_G^{\gamma/2}\{S_F^{\gamma/2}\{S_F^{\gamma/2}[S_G^{\gamma/2}(y_0)]\}\}.$$

From the previous estimation of Lie–Trotter splitting scheme we have

$$S_{F+G}^{\gamma/2}(y_0) - S_G^\gamma[S_F^\gamma(y_0)] = \frac{\gamma^2}{8}\{F'(y_0)G(y_0) - G'(y_0)F(y_0)\} + O(\gamma^3),$$

$$S_{F+G}^{\gamma/2}(y_0) - S_F^\gamma[S_G^\gamma(y_0)] = \frac{\gamma^2}{8}\{G'(y_0)F(y_0) - F'(y_0)G(y_0)\} + O(\gamma^3).$$

---

[4]Since a time-dependent ODE can be formulated as a time-independent ODE by introducing an auxiliary variable (Chicone, 2007), the theorem here developed for time-independent ODEs can also be applied to time-dependent ODEs without loss of generality.

Combine the two estimations, we have

$$
\begin{aligned}
S_G^{\gamma/2}\{S_F^{\gamma/2}\{S_F^{\gamma/2}[S_G^{\gamma/2}(y_0)]\}\} = \; & S_{F+G}^{\gamma}(y_0) + \frac{\gamma^2}{8}\{F'(y_0)G(y_0) - G'(y_0)F(y_0)\} \\
& + \frac{\gamma^2}{8}\{G'(y_0)F(y_0) - F'(y_0)G(y_0)\} + O(\gamma^3) \\
= \; & S_{F+G}^{\gamma}(y_0) + O(\gamma^3).
\end{aligned}
$$

$\square$

## B  EXPERIMENT SETTINGS

### B.1  MACHINE TRANSLATION

**Dataset**  The training/validation/test sets of the IWSLT14 dataset contain about 153K/7K/7K sentence pairs, respectively. We use a vocabulary of 10K tokens based on a joint source and target byte pair encoding (BPE) (Sennrich et al., 2016). For WMT14 dataset, we replicate the setup of Vaswani et al. (2017), which contains 4.5M training parallel sentence pairs. Newstest2014 is used as the test set, and Newstest2013 is used as the validation set. The 37K vocabulary for WMT14 is based on a joint source and target BPE factorization.

**Model**  For the WMT14 dataset, the basic configurations of the Transformer architecture are the `base` and the `big` settings (Vaswani et al., 2017). Both of them consist of a 6-layer encoder and 6-layer decoder. The size of the hidden nodes and embeddings are set to 512 for `base` and 1024 for `big`. The number of heads are 8 for `base` and 16 for `big`. Since the IWSLT14 dataset is much smaller than the WMT14 dataset, the `small` setting is usually used, whose size of hidden states and embeddings is set to 512 and the number of heads is set to 4. For all settings, the dimensionality of the inner-layer of the position-wise FFN is four times of the dimensionality of the hidden states.

For each setting (`base`, `big` and `small`), we replace all Transformer layers by the Macaron layers and obtain the `base`, `big` and `small` Macaron, each of which contains two position-wise feed-forward sub-layers in a layer. The translation model is based on the encoder-decoder framework. In the Transformer, the decoder layer has a third sub-layer which performs multi-head attention over the output of the encoder stack (encoder-decoder-attention) and a mask to prevent positions from attending to subsequent positions. In our implementation of Macaron decoder, we also use masks and split the FFN into two sub-layers and thus our decoder layer is (FFN, self-attention, encoder-decoder-attention and FFN).

To make a fair comparison, we set the dimensionality of the inner-layer of the two FFN sub-layers in the Macaron layers to two times of the dimensionality of the hidden states. By doing this, the `base`, `big` and `small` Macaron have the same number of parameters as the `base`, `big` and `small` Transformer respectively.

**Optimizer and training**  We use the Adam optimizer and follow the optimizer setting and learning rate schedule in Vaswani et al. (2017). For the `big` setting, we enlarge the batch size and learning rate as suggested in Ott et al. (2018) to accelerate training. We employ label smoothing of value $\epsilon_{ls} = 0.1$ (Szegedy et al., 2016) in all experiments. Models for WMT14/IWSLT14 are trained on 4/1 NVIDIA P40 GPUs respectively. Our code is based on the open-sourced `fairseq` (Gehring et al., 2017) code base in PyTorch toolkit.

**Evaluation**  We use BLEU[5] (Papineni et al., 2002) as the evaluation measure for machine translation. Following common practice, we use tokenized case-sensitive BLEU and case-insensitive BLEU for WMT14 En-De and IWSLT14 De-En respectively. During inference, we use beam search with beam size 4 and length penalty 0.6 for WMT14, and beam size 5 and length penalty 1.0 for IWSLT14, following Vaswani et al. (2017).

---

[5]`https://github.com/moses-smt/mosesdecoder/blob/master/scripts/generic/multi-bleu.perl`

## B.2 Unsupervised Pretraining

**Pre-training dataset**   We follow Devlin et al. (2018) to use English Wikipedia corpus and Book-Corpus for pre-training. As the dataset BookCorpus (Zhu et al., 2015) is no longer freely distributed. We follow the suggestions from Devlin et al. (2018) to crawl and collect BookCorpus[6] on our own. The concatenation of two datasets includes roughly 3.4B words in total, which is comparable with the data corpus used in Devlin et al. (2018). We first segment documents into sentences with Spacy;[7] Then, we normalize, lower-case, and tokenize texts using Moses (Koehn et al., 2007) and apply BPE (Sennrich et al., 2016). We randomly split documents into one training set and one validation set. The training-validation ratio for pre-training is 199:1.

**Model**   We compare our proposed Macaron Net with the `base` setting from the original paper (Devlin et al., 2018), which consists of 12 Transformer layers. The size of hidden states and embeddings are set to 768, and the number of attention heads is set to 12. Similarly, we replace the Transformer layers in BERT `base` by the Macaron layers and reduce the dimensionality of the inner-layer of the two FFN sub-layers by half, and thus we keep the number of parameters of our Macaron `base` as the same as BERT `base`.

**Optimizer and training**   We follow Devlin et al. (2018) to use two tasks to pretrain our model. One task is masked language modeling, which masks some percentage of the input tokens at random, and then requires the model to predict those masked tokens. Another task is next sentence prediction, which requires the model to predict whether two sentences in a given sentence pair are consecutive. We use the Adam optimizer and follow the optimizer setting and learning rate schedule in Devlin et al. (2018) and trained the model on 4 NVIDIA P40 GPUs.

## B.3 GLUE Dataset

We provide a brief description of the tasks in the GLUE benchmark (Wang et al., 2019) and our fine-tuning process on the GLUE datasets.

**CoLA**   The Corpus of Linguistic Acceptability (Warstadt et al., 2018) consists of English acceptability judgments drawn from books and journal articles on linguistic theory. The task is to predict whether an example is a grammatical English sentence. The performance is evaluated by Matthews correlation coefficient (Matthews, 1975).

**SST-2**   The Stanford Sentiment Treebank (Socher et al., 2013) consists of sentences from movie reviews and human annotations of their sentiment. The task is to predict the sentiment of a given sentence (positive/negative). The performance is evaluated by the test accuracy.

**MRPC**   The Microsoft Research Paraphrase Corpus (Dolan & Brockett, 2005) is a corpus of sentence pairs automatically extracted from online news sources, with human annotations for whether the sentences in the pair are semantically equivalent, and the task is to predict the equivalence. The performance is evaluated by both the test accuracy and the test F1.

**STS-B**   The Semantic Textual Similarity Benchmark (Cer et al., 2017) is a collection of sentence pairs drawn from news headlines, video and image captions, and natural language inference data. Each pair is human-annotated with a similarity score from 1 to 5; the task is to predict these scores. The performance is evaluated by Pearson and Spearman correlation coefficients.

**QQP**   The Quora Question Pairs[8] (Chen et al., 2018b) dataset is a collection of question pairs from the community question-answering website Quora. The task is to determine whether a pair of questions are semantically equivalent. The performance is evaluated by both the test accuracy and the test F1.

---

[6] `https://www.smashwords.com/`
[7] `https://spacy.io`
[8] `https://data.quora.com/First-Quora-Dataset-Release-Question-Pairs`

**MNLI**    The Multi-Genre Natural Language Inference Corpus (Williams et al., 2018) is a crowd-sourced collection of sentence pairs with textual entailment annotations. Given a premise sentence and a hypothesis sentence, the task is to predict whether the premise entails the hypothesis (*entailment*), contradicts the hypothesis (*contradiction*), or neither (*neutral*). The performance is evaluated by the test accuracy on both *matched* (in-domain) and *mismatched* (cross-domain) sections of the test data.

**QNLI**    The Question-answering NLI dataset is converted from the Stanford Question Answering Dataset (SQuAD) (Rajpurkar et al., 2016) to a classification task. The performance is evaluated by the test accuracy.

**RTE**    The Recognizing Textual Entailment (RTE) datasets come from a series of annual textual entailment challenges (Bentivogli et al., 2009). The task is to predict whether sentences in a sentence pair are entailment. The performance is evaluated by the test accuracy.

**WNLI**    The Winograd Schema Challenge (Levesque et al., 2011) is a reading comprehension task in which a system must read a sentence with a pronoun and select the referent of that pronoun from a list of choices. We follow Devlin et al. (2018) to skip this task in our experiments because few previous works do better than predicting the majority class for this task.

**Fine-tuning on GLUE tasks**    To fine-tune the models, following Devlin et al. (2018), we search the optimization hyperparameters in a search space including different batch sizes (16/32), learning rates (5e-3/3e-5), number of epochs (3/4/5), and a set of different random seeds. We select the model for testing according to their performance on the development set.

**Test data**    Note that the GLUE dataset distribution does not include the Test labels, and we only made a single GLUE evaluation server submission[9] for each of our models.

---

[9] https://gluebenchmark.com

