# OpenReview forum: "Understanding and Improving Transformer From a Multi-Particle Dynamic System Point of View"
_ICLR.cc/2020/Conference — Reject_

### Official Review · AnonReviewer1 · 2019-10-14
**Official Blind Review #1**

**Rating:** 1

**Review:**

Contributions: This paper builds an ad-hoc connection between the Transformer and the numerical ODE solver (the Lie-Trotter splitting scheme and the Euler's method) for a convection-diffusion equation in a multi-particle dynamic system. Then, the author(s) developed an ad-hoc Strang-Marchuk splitting style architecture, named Macaron Net. Finally, this paper provides some experiments to verify the performance of the proposed architecture. However, the comparisons with the benchmark results are questionable. I have listed my concerns in the Experiment section.


Motivation: This paper developed the Macaron Net based on a locally third-order operator splitting scheme for the convection-diffusion equation. However, there is no theoretical interpretation of why third-order splitting corresponding to better architecture. Theorem 1 in the paper is a known result, and it is irrelevant to the paper, I highly recommend the author to remove it from the main text. I also suggest the author explore more operator splitting schemes and do a systematic comparison between them. Moreover, I think it will be a real contribution if the author can analyze the error between the numerical scheme and architectures.


Reformulate Transformer Layers ans an ODE solver for Multi-Particle Dynamic System: There is a big gap between Eqns 3, 4 and 5. Why F represents a diffusion term, why G represents a convection term? From a statistical mechanics point of view, this comparison does not make sense. I do not buy this model.


Related Work: There is no related-work section that discusses the related work, and all the referenced papers are generic. For instance, the efforts in developing language models, the application of convection-diffusion equation, and String-Marchuck and other operator splitting schemes in machine learning. The author should better position the paper to exist work.


Experiments: This section is extremely questionable. My initial thought after reading the reported results is that the architecture proposed in this paper easily outperforms the existing work. However, after I do a cross-check with the existing work, I found the author did not compare with the best results reported in the benchmark work and hide much information. After simply checking two existing papers, I found that the author ignored the comparison with BERT large. Also, the author ignored the most important result reported by Wu et al. 2019b.  To be fair, the author should perform an apple-to-apple comparison with the existing work and report the uncertainties in their results. Moreover, the author should report the parameters used in all their experiments.


I think this heuristic study might be a contribution to ICLR if all my concerns are addressed, and I am willing to raise my rating to accept.

**Experience Assessment:**

I have published in this field for several years.

**Review Assessment: Checking Correctness Of Derivations And Theory:**

I carefully checked the derivations and theory.

**Review Assessment: Checking Correctness Of Experiments:**

I carefully checked the experiments.

**Review Assessment: Thoroughness In Paper Reading:**

I read the paper thoroughly.

---

### Official Review · AnonReviewer3 · 2019-10-16
**Official Blind Review #3**

**Rating:** 3

**Review:**

In this work, the authors show that the sequence of self-attention and feed-forward layers within a Transformer can be interpreted as an approximate numerical solution to a set of coupled ODEs. Based on this insight, the authors propose to replace the first-order Lie-Trotter splitting scheme by the more accurate, second-order Strang splitting scheme. They then present experimental results that indicate an improved performance of their Macaron Net compared to the Transformer and argue that this is due to the former being a more accurate numerical solution to the underlying set of ODEs.

The authors highlight an interesting connection between the Transformer architecture and ODEs. In particular, they derive a set of ODEs that is solved numerically by the Transformer and borrow from the body of literature on numerical ODE solvers to improve the architecture. I find that this is a very elegant and promising approach for finding better architectures.

However, I also identified two major and a couple of minor shortcomings of the paper that are explained in detail below. Based on these shortcomings, I recommend rejecting the paper but I would be willing to increase the score if these points were addressed in sufficient detail.

Major points:

1) Replacing the first-order operator splitting scheme by a second-order scheme only guarantees a lower overall truncation error if the split ODEs are solved with sufficiently high accuracy. In particular, the overall accuracy of the numerical solution to the original ODE depends on the accuracy of the operator splitting and the accuracy of the integration scheme used to solve the split ODEs (e.g. Euler’s method). The authors improve the operator splitting, i.e. they make it second-order, but they keep Euler’s method to integrate the individual ODEs. Because of that, they actually do not get rid of the lowest-order error term of the overall scheme and therefore do not obtain a more accurate ODE solver. I think this is a crucial point that invalidates the authors’ claim that the Macaron Net employs a higher-order integration scheme. As far as I am aware, this is not commented on in the paper at all. To address this shortcoming, the authors could replace Euler’s method by a second-order integrator.

2) The experiments considered in this paper are interesting and show competitive performance but, in my opinion, they do not sufficiently support the claim that the Macaron Net yields a more accurate solution to the underlying set of ODEs compared to a Transformer. For a more convincing support of this claim, the authors could consider a toy problem, i.e. a simple set of ODEs with known analytical solution, and actually show that the Macaron Net is more accurate. The accuracy of a numerical ODE solver is commonly assessed by plotting the absolute difference between the exact solution (or a high-resolution numerical approximation to it) and the numerical solution vs the timestep (here \gamma). I suspect that such an analysis would support my previous comment and show that the proposed new architecture is not more accurate ODE solver than the original one.

Minor points and questions:

i) Eqs. (17-19) suggest that you apply two different FFN layers (doubling the number of parameters) instead of applying the same FFN layer twice. You comment on this in Sec. 4.1 when you say ‘..., we set the dimensionality of the inner-layer of the two FFN sub-layers in the Macaron layers to two times of the dimensionality of the hidden states’. It is not clear to me why consistency with Strang splitting requires two different layers rather than applying the same FFN layer twice. Is the reason for having a separate, trainable layer to account for the explicit time dependence of G in Eq. (16)? I think that this is a very important point that should be clarified.

ii) I think that this type of system is usually referred to as ‘dynamical system’ and not ‘dynamic system’. Please check that and, if applicable, update the title.

iii) The authors say that Eq. (5) is a 'convection-diffusion equation’. As far as I am aware, the diffusion equation is a partial differential equation (PDE). Perhaps there is a different notion 'diffusion equation’ in ODE theory. If that’s the case, could the authors please clarify this point to avoid confusion, e.g. by adding a suitable reference in which this type of ODE is classified as a convection-diffusion equation?

iv) In Sec. 2 (2nd paragraph),  you cite Vaswani et al. (2017) but in that work the quantity under the square-root in the denominator of Attention(Q, K, V) is actually d_k, the dimension of the key, and not d_model.

v) Figure 1 is a very vague illustration of the connection to ODEs and provides almost no explanation in the caption. I don’t think there is much value in having this figure there.

vi) There are a couple of mistakes in the paper (grammar and expressions) that should be fixed. For example, ‘the Euler’s method’ instead of ‘Euler’s method’, ‘movement in the space’ instead of ‘movement in space’, ‘dynamic system’ instead of ‘dynamical system’, ‘project parameter matrices’ instead of ‘parameter matrices’ or ‘projections’, ‘specially’ instead of ‘specifically’, etc. Please take a look at the relevant sections in the paper and revise them accordingly.

vii) You explain multiple times why the proposed architecture is called a ‘Macaron Net’ (Abstract, Sec 1, Sec. 3). To avoid repetition, I would only explain it once.


**Experience Assessment:**

I have published one or two papers in this area.

**Review Assessment: Checking Correctness Of Derivations And Theory:**

I assessed the sensibility of the derivations and theory.

**Review Assessment: Checking Correctness Of Experiments:**

I assessed the sensibility of the experiments.

**Review Assessment: Thoroughness In Paper Reading:**

I read the paper at least twice and used my best judgement in assessing the paper.

---

### Official Review · AnonReviewer2 · 2019-10-21
**Official Blind Review #2**

**Rating:** 3

**Review:**

The paper points out a formal analogy between transformers and an ODE modelling multi-particle convection (the feed-forward network) and diffusion (the self-attention head). The paper then adapts the Strang-Marchuk splitting scheme for solving ODEs to construct a slightly different transformer architecture: “FFN of Attention of FFN”, instead of “FFN of Attention”. The new architecture, refered to as a Macaron-Net, yields better performance in a variety of experiments.

PROs
1. The proposed new architecture is fairly simple.
2. The experimental results are fairly good.

CONs
1. Introducing two feedforward layers with *different* parameters W^{down} and W^{up} is a significant deviation from Strang-Marchuk splitting. I expected the two FFNs inside the Macaron to have the same weights. As I understand it, the motivation for the splitting is to improve the numerical performance of the update scheme for the ODE. In contrast, allowing different weights for the FFNs means the “physical process” is now a lot more “free”. Is there any “physical” motivation for the different parameters? (Beyond the fact that it improves performance). How much worse is empirical performance when the parameters are the same?

2. Following on from the above point, the analogy between the multi-particle system and the transformer is quite weak. The equations look similar when you squint the right way. But that’s as far as it goes. Fig 1 is a nice visualization, but it doesn’t provide insight into the dynamics of transformers. What does “Particles move in the space along time (Semantics encoded in stacked neural network layers)” mean? How do the particles connect to the semantics?
3. The proof of Bobylev & Ohwada’s theorem is included in the paper. Is there any connection between the theorem (or the techniques used in its proof) and transformers? I suspect the answer is no.

SUMMARY
In short, the paper (i) proposes two FFN layers instead of one in each block of the transformer and (ii) shows it performs slightly better than before. This is decent, but in my opinion not enough the clear the bar for ICLR.

The connection to multi-particle ODEs is genuinely interesting. However, it is not sufficiently fleshed out to count as a contribution (yet). It’s possible the authors have discovered something deep. It’s also possible they got lucky with a physically motivated modification of transformers that actually has nothing to do with the dynamics of multi-particle systems. I’m not sure what further experiments would be needed to make the case. But I recommend the authors dig into the equations and the dynamics to see what it really going on under the hood. Just showing improved performance on a few benchmarks is not enough to convince the connection is solid.


**Experience Assessment:**

I have read many papers in this area.

**Review Assessment: Checking Correctness Of Derivations And Theory:**

I carefully checked the derivations and theory.

**Review Assessment: Checking Correctness Of Experiments:**

I assessed the sensibility of the experiments.

**Review Assessment: Thoroughness In Paper Reading:**

I read the paper thoroughly.

---

### Author Response · Authors · 2019-11-15
**Author Response**

We thank all reviewers for the valuable comments.  Since the concerns are shared, we decide to answer all of the questions here.

[Regarding Strang-Marchuk splitting]

First, Strang-Marchuk splitting can also be used for  nonautonomous system (right hand side is varying from time, i.e. \dot x = f(x,t) but not \dot x = f(x)). In fact the nonautonomous system and autonomous system are nearly the same, for you can write down a nonautonomous system $\dot x = f(x,t)$ to an equivalent autonomous system  $d [x, R]^T/dt = [f(x,R),1]^T$.  The two FFN blocks have different weights can also be considered as Strang splitting. Moreover, applying the sub-layer twice will make the optimization landscape worse.

Second, the numerical error is a composition of the splitting scheme error and the integration method. We agree it’s better to replace the Euler's method with a higher-order one and build a higher-order better algorithm. However, such changing is expensive. That’s why we are aiming to have a better splitting, which also helps.

[Regarding the diffusion and convection]

G represents a convection term is due to that the particles are moving in their own ways. In a mean-field viewpoint, the distribution of the particles will move like a transport equation $\partial \rho=\nabla\cdot(\rho G)$.
The derivation of diffusion PDEs is from a particle system, i.e., limiting the number of particle numbers to infinity(mean-field limit), the evolution of the probability measure will come to a diffusion-like process. We also want to point out that the attention network is not a standard diffusion, it’s more like a nonlocal diffusion as proposed in the following paper.

Tao Y, Sun Q, Du Q, et al. Nonlocal neural networks, nonlocal diffusion and nonlocal modeling[C]//Advances in Neural Information Processing Systems. 2018: 496-506.

---

### Decision · Program_Chairs · 2019-12-19

**Decision:**

Reject

**Comment:**

In this work, the authors interpret the Transformer as a numerical ODE modelling multi-particle convection. Guided by this connection, the authors take the Transformer that uses a feed forward net over attentions, and create a variant of transformer which instead uses an FFN-attention-FFN layer, thus the name macaron net. The authors present experiments in the GLUE dataset and in two MT datasets, and they overall report improved performance using their variant of Transformer. Thus, the main selling point of the paper is how seeing Transformer under his new light can potentially improve results through the construction of better models. The main criticisms from the authors is that  this story is not entirely convincing because the proposed variant departs a bit from the theory (R1 and comment about the Strang-Marchuk splitting) and the papers does not consider an evaluation of accuracy of Macaron in solving the underlying set of ODEs (comment from R3). As such, I cannot recommend acceptance of this paper -- I believe another set of revisions would increase the impact of this paper.